# Optimal Black-Box Reductions Between Optimization Objectives[*]

**Zeyuan Allen-Zhu**
zeyuan@csail.mit.edu
Institute for Advanced Study
& Princeton University

**Elad Hazan**
ehazan@cs.princeton.edu
Princeton University

## Abstract

The diverse world of machine learning applications has given rise to a plethora of algorithms and optimization methods, finely tuned to the specific regression or classification task at hand. We reduce the complexity of algorithm design for machine learning by reductions: we develop reductions that take a method developed for one setting and apply it to the entire spectrum of smoothness and strong-convexity in applications.

Furthermore, unlike existing results, our new reductions are *optimal* and more *practical*. We show how these new reductions give rise to new and faster running times on training linear classifiers for various families of loss functions, and conclude with experiments showing their successes also in practice.

## 1 Introduction

The basic machine learning problem of minimizing a regularizer plus a loss function comes in numerous different variations and names. Examples include Ridge Regression, Lasso, Support Vector Machine (SVM), Logistic Regression and many others. A multitude of optimization methods were introduced for these problems, but in most cases specialized to very particular problem settings. Such specializations appear necessary since objective functions for different classification and regularization tasks admin different convexity and smoothness parameters. We list below a few recent algorithms along with their applicable settings.

- Variance-reduction methods such as SAGA and SVRG [9, 14] *intrinsically require* the objective to be *smooth*, and do not work for non-smooth problems like SVM. This is because for loss functions such as hinge loss, no unbiased gradient estimator can achieve a variance that approaches to zero.

- Dual methods such as SDCA or APCG [20, 30] *intrinsically require* the objective to be *strongly convex (SC)*, and do not directly apply to non-SC problems. This is because for a non-SC objective such as Lasso, its dual is not well-behaved due to the $\ell_1$ regularizer.

- Primal-dual methods such as SPDC [35] require the objective to be both smooth and SC. Many other algorithms are only analyzed for both smooth and SC objectives [7, 16, 17].

In this paper we investigate whether such specializations are inherent. Is it possible to take a convex optimization algorithm designed for one problem, and apply it to different classification or regression settings in a black-box manner? Such a reduction should ideally take full and *optimal* advantage of the objective properties, namely strong-convexity and smoothness, for each setting.

Unfortunately, existing reductions are still very limited for at least two reasons. First, they incur at least a logarithmic factor $\log(1/\varepsilon)$ in the running time so leading only to suboptimal convergence

---

[*]The full version of this paper can be found on https://arxiv.org/abs/1603.05642. This paper is partially supported by an NSF Grant, no. 1523815, and a Microsoft Research Grant, no. 0518584.

rates.[2] Second, after applying existing reductions, algorithms become *biased* so the objective value does not converge to the global minimum. These theoretical concerns also translate into running time losses and parameter tuning difficulties in practice.

In this paper, we develop new and *optimal* regularization and smoothing reductions that can

- shave off a non-optimal $\log(1/\varepsilon)$ factor
- produce unbiased algorithms

Besides such technical advantages, our new reductions also enable researchers to focus on designing algorithms for only one setting but infer optimal results more broadly. This is opposed to results such as [4, 25] where the authors develop *ad hoc* techniques to tweak specific algorithms, rather than all algorithms, and apply them to other settings without losing extra factors and without introducing bias.

Our new reductions also enable researchers to prove *lower bounds* more broadly [32].

## 1.1 Formal Setting and Classical Approaches

Consider minimizing a composite objective function

$$\min_{x \in \mathbb{R}^d} \left\{ F(x) \stackrel{\text{def}}{=} f(x) + \psi(x) \right\} \ , \tag{1.1}$$

where $f(x)$ is a differentiable convex function and $\psi(x)$ is a relatively simple (but possibly non-differentiable) convex function, sometimes referred to as the *proximal* function. Our goal is to find a point $x \in \mathbb{R}^d$ satisfying $F(x) \le F(x^*) + \varepsilon$, where $x^*$ is a minimizer of $F$.

In most classification and regression problems, $f(x)$ can be written as $f(x) = \frac{1}{n} \sum_{i=1}^n f_i(\langle x, a_i \rangle)$ where each $a_i \in \mathbb{R}^d$ is a feature vector. We refer to this as the *finite-sum* case of (1.1).

- CLASSICAL REGULARIZATION REDUCTION.

  Given a non-SC $F(x)$, one can define a new objective $F'(x) \stackrel{\text{def}}{=} F(x) + \frac{\sigma}{2}\|x_0 - x\|^2$ in which $\sigma$ is on the order of $\varepsilon$. In order to minimize $F(x)$, the classical regularization reduction calls an oracle algorithm to minimize $F'(x)$ instead, and this oracle only needs to work with SC functions.

  EXAMPLE. If $F$ is $L$-smooth, one can apply accelerated gradient descent to minimize $F'$ and obtain an algorithm that converges in $O(\sqrt{L/\varepsilon} \log \frac{1}{\varepsilon})$ iterations in terms of minimizing the original $F$. This complexity has a suboptimal dependence on $\varepsilon$ and shall be improved using our new regularization reduction.

- CLASSICAL SMOOTHING REDUCTION (FINITE-SUM CASE).[3]

  Given a non-smooth $F(x)$ of a finite-sum form, one can define a smoothed variant $\widehat{f}_i(\alpha)$ for each $f_i(\alpha)$ and let $F'(x) = \frac{1}{n} \sum_{i=1}^n \widehat{f}_i(\langle a_i, x \rangle) + \psi(x)$. [4] In order to minimize $F(x)$, the classical smoothing reduction calls an oracle algorithm to minimize $F'(x)$ instead, and this oracle only needs to work with smooth functions.

  EXAMPLE. If $F(x)$ is $\sigma$-SC and one applies accelerated gradient descent to minimize $F'$, this yields an algorithm that converges in $O\left(\frac{1}{\sqrt{\sigma \varepsilon}} \log \frac{1}{\varepsilon}\right)$ iterations for minimizing the original $F(x)$. Again, the additional factor $\log(1/\varepsilon)$ can be removed using our new smoothing reduction.

Besides the non-optimality, applying the above two reductions gives only *biased* algorithms. One has to tune the regularization or smoothing parameter, and the algorithm only converges to the minimum of the regularized or smoothed problem $F'(x)$, which can be away from the true minimizer of $F(x)$ by a distance proportional to the parameter. This makes the reduction hard to use in practice.

## 1.2 Our New Results

To introduce our new reductions, we first define a property on the oracle algorithm.

**Our Black-Box Oracle.** Consider an algorithm $\mathcal{A}$ that minimizes (1.1) when the objective $F$ is $L$-smooth and $\sigma$-SC. We say that $\mathcal{A}$ satisfies the *homogenous objective decrease* (HOOD) property in time $\mathsf{Time}(L, \sigma)$ if, for every starting vector $x_0$, $\mathcal{A}$ produces an output $x'$ satisfying $F(x') - F(x^*) \leq \frac{F(x_0) - F(x^*)}{4}$ in time $\mathsf{Time}(L, \sigma)$. In other words, $\mathcal{A}$ decreases the objective value distance to the minimum by a constant factor in time $\mathsf{Time}(L, \sigma)$, regardless of how large or small $F(x_0) - F(x^*)$ is. We give a few example algorithms that satisfy HOOD:

- Gradient descent and accelerated gradient descent satisfy HOOD with $\mathsf{Time}(L, \sigma) = O(L/\sigma) \cdot C$ and $\mathsf{Time}(L, \sigma) = O(\sqrt{L/\sigma}) \cdot C$ respectively, where $C$ is the time needed to compute a gradient $\nabla f(x)$ and perform a proximal gradient update [23]. Many subsequent works in this line of research also satisfy HOOD, including [3, 7, 16, 17].

- SVRG and SAGA [14, 34] solve the finite-sum form of (1.1) and satisfy HOOD with $\mathsf{Time}(L, \sigma) = O(n + L/\sigma) \cdot C_1$ where $C_1$ is the time needed to compute a stochastic gradient $\nabla f_i(x)$ and perform a proximal gradient update.

- Katyusha [1] solves the finite-sum form of (1.1) and satisfies HOOD with $\mathsf{Time}(L, \sigma) = O(n + \sqrt{nL/\sigma}) \cdot C_1$.

**AdaptReg.** For objectives $F(x)$ that are non-SC and $L$-smooth, our AdaptReg reduction calls the an oracle satisfying HOOD a logarithmic number of times, each time with a SC objective $F(x) + \frac{\sigma}{2}\|x - x_0\|^2$ for an exponentially decreasing value $\sigma$. In the end, AdaptReg produces an output $\widehat{x}$ satisfying $F(\widehat{x}) - F(x^*) \leq \varepsilon$ with a total running time $\sum_{t=0}^{\infty} \mathsf{Time}(L, \varepsilon \cdot 2^t)$.

Since most algorithms have an inverse polynomial dependence on $\sigma$ in $\mathsf{Time}(L, \sigma)$, when summing up $\mathsf{Time}(L, \varepsilon \cdot 2^t)$ for positive values $t$, we do not incur the additional factor $\log(1/\varepsilon)$ as opposed to the old reduction. In addition, AdaptReg is an *unbiased* and *anytime* algorithm. $F(\widehat{x})$ converges to $F(x^*)$ as the time goes without the necessity of changing parameters, so the algorithm can be interrupted at any time. We mention some theoretical applications of AdaptReg:

- Applying AdaptReg to SVRG, we obtain a running time $O(n \log \frac{1}{\varepsilon} + \frac{L}{\varepsilon}) \cdot C_1$ for minimizing finite-sum, non-SC, and smooth objectives (such as Lasso and Logistic Regression). This improves on known theoretical running time obtained by non-accelerated methods, including $O(n \log \frac{1}{\varepsilon} + \frac{L}{\varepsilon} \log \frac{1}{\varepsilon}) \cdot C_1$ through the old reduction, as well as $O(\frac{n+L}{\varepsilon}) \cdot C_1$ through direct methods such as SAGA [9] and SAG [27].

- Applying AdaptReg to Katyusha, we obtain a running time $O(n \log \frac{1}{\varepsilon} + \frac{\sqrt{nL}}{\sqrt{\varepsilon}}) \cdot C_1$ for minimizing finite-sum, non-SC, and smooth objectives (such as Lasso and Logistic Regression). This is the first and only known stochastic method that converges with the optimal $1/\sqrt{\varepsilon}$ rate (as opposed to $\log(1/\varepsilon)/\sqrt{\varepsilon}$) for this class of objectives. [1]

- Applying AdaptReg to methods that do not originally work for non-SC objectives such as [7, 16, 17], we improve their running times by a factor of $\log(1/\varepsilon)$ for working with non-SC objectives.

**AdaptSmooth and JointAdaptRegSmooth.** For objectives $F(x)$ that are finite-sum, $\sigma$-SC, but non-smooth, our AdaptSmooth reduction calls an oracle satisfying HOOD a logarithmic number of times, each time with a smoothed variant of $F^{(\lambda)}(x)$ and an exponentially decreasing smoothing parameter $\lambda$. In the end, AdaptSmooth produces an output $\widehat{x}$ satisfying $F(\widehat{x}) - F(x^*) \leq \varepsilon$ with a total running time $\sum_{t=0}^{\infty} \mathsf{Time}(\frac{1}{\varepsilon \cdot 2^t}, \sigma)$.

Since most algorithms have a polynomial dependence on $L$ in $\mathsf{Time}(L, \sigma)$, when summing up $\mathsf{Time}(\frac{1}{\varepsilon \cdot 2^t}, \sigma)$ for positive values $t$, we do not incur an additional factor of $\log(1/\varepsilon)$ as opposed to the old reduction. AdaptSmooth is also an *unbiased* and *anytime* algorithm for the same reason as AdaptReg.

In addition, AdaptReg and AdaptSmooth can effectively work together, to solve finite-sum, non-SC, and non-smooth case of (1.1), and we call this reduction JointAdaptRegSmooth.

We mention some theoretical applications of AdaptSmooth and JointAdaptRegSmooth:

- Applying AdaptReg to `Katyusha`, we obtain a running time $O\big(n\log\frac{1}{\varepsilon}+\frac{\sqrt{n}}{\sqrt{\sigma\varepsilon}}\big)\cdot C_1$ for minimizing finite-sum, SC, and non-smooth objectives (such as SVM). Therefore, `Katyusha` combined with AdaptReg is the first and only known stochastic method that converges with the optimal $1/\sqrt{\varepsilon}$ rate (as opposed to $\log(1/\varepsilon)/\sqrt{\varepsilon}$) for this class of objectives. [1]

- Applying JointAdaptRegSmooth to `Katyusha`, we obtain a running time $O\big(n\log\frac{1}{\varepsilon}+\frac{\sqrt{n}}{\varepsilon}\big)\cdot C_1$ for minimizing finite-sum, SC, and non-smooth objectives (such as L1-SVM). Therefore, `Katyusha` combined with JointAdaptRegSmooth gives the first and only known stochastic method that converges with the optimal $1/\varepsilon$ rate (as opposed to $\log(1/\varepsilon)/\varepsilon$) for this class of objectives. [1]

**Roadmap.** We provide notations in Section 2 and discuss related works in Section 3. We propose AdaptReg in Section 4 and AdaptSmooth in Section 5. We leave proofs as well as the description and analysis of JointAdaptRegSmooth to the full version of this paper. We include experimental results in Section 7.

## 2 Preliminaries

In this paper we denote by $\nabla f(x)$ the full gradient of $f$ if it is differentiable, or the subgradient if $f$ is only Lipschitz continuous. Recall some classical definitions on strong convexity and smoothness.

**Definition 2.1** (smoothness and strong convexity). *For a convex function $f\colon\mathbb{R}^n\to\mathbb{R}$,*

- *$f$ is $\sigma$-strongly convex if $\forall x,y\in\mathbb{R}^n$, it satisfies $f(y)\geq f(x)+\langle\nabla f(x),y-x\rangle+\frac{\sigma}{2}\|x-y\|^2$.*
- *$f$ is $L$-smooth if $\forall x,y\in\mathbb{R}^n$, it satisfies $\|\nabla f(x)-\nabla f(y)\|\leq L\|x-y\|$.*

**Characterization of SC and Smooth Regimes.** In this paper we give numbers to the following 4 categories of objectives $F(x)$ in (1.1). Each of them corresponds to some well-known training problems in machine learning. (Letting $(a_i,b_i)\in\mathbb{R}^d\times\mathbb{R}$ be the $i$-th feature vector and label.)

Case 1: $\psi(x)$ is $\sigma$-SC and $f(x)$ is $L$-smooth. Examples:
- *ridge regression*: $f(x)=\frac{1}{2n}\sum_{i=1}^n(\langle a_i,x\rangle-b_i)^2$ and $\psi(x)=\frac{\sigma}{2}\|x\|_2^2$.
- *elastic net*: $f(x)=\frac{1}{2n}\sum_{i=1}^n(\langle a_i,x\rangle-b_i)^2$ and $\psi(x)=\frac{\sigma}{2}\|x\|_2^2+\lambda\|x\|_1$.

Case 2: $\psi(x)$ is non-SC and $f(x)$ is $L$-smooth. Examples:
- *Lasso*: $f(x)=\frac{1}{2n}\sum_{i=1}^n(\langle a_i,x\rangle-b_i)^2$ and $\psi(x)=\lambda\|x\|_1$.
- *logistic regression*: $f(x)=\frac{1}{n}\sum_{i=1}^n\log(1+\exp(-b_i\langle a_i,x\rangle))$ and $\psi(x)=\lambda\|x\|_1$.

Case 3: $\psi(x)$ is $\sigma$-SC and $f(x)$ is non-smooth (but Lipschitz continuous). Examples:
- *SVM*: $f(x)=\frac{1}{n}\sum_{i=1}^n\max\{0,1-b_i\langle a_i,x\rangle\}$ and $\psi(x)=\sigma\|x\|_2^2$.

Case 4: $\psi(x)$ is non-SC and $f(x)$ is non-smooth (but Lipschitz continuous). Examples:
- *$\ell_1$-SVM*: $f(x)=\frac{1}{n}\sum_{i=1}^n\max\{0,1-b_i\langle a_i,x\rangle\}$ and $\psi(x)=\lambda\|x\|_1$.

**Definition 2.2** (HOOD property). *We say an algorithm $\mathcal{A}(F,x_0)$ solving Case 1 of problem (1.1) satisfies the* homogenous objective decrease *(HOOD) property with time $\mathsf{Time}(L,\sigma)$ if, for every starting point $x_0$, it produces output $x'\leftarrow\mathcal{A}(F,x_0)$ such that $F(x')-\min_x F(x)\leq\frac{F(x_0)-\min_x F(x)}{4}$ in time $\mathsf{Time}(L,\sigma)$.*[5]

In this paper, we denote by $C$ the time needed for computing a full gradient $\nabla f(x)$ and performing a proximal gradient update of the form $x'\leftarrow\arg\min_x\big\{\frac{1}{2}\|x-x_0\|^2+\eta(\langle\nabla f(x),x-x_0\rangle+\psi(x))\big\}$. For the finite-sum case of problem (1.1), we denote by $C_1$ the time needed for computing a stochastic (sub-)gradient $\nabla f_i(\langle a_i,x\rangle)$ and performing a proximal gradient update of the form $x'\leftarrow\arg\min_x\big\{\frac{1}{2}\|x-x_0\|^2+\eta(\langle\nabla f_i(\langle a_i,x\rangle)a_i,x-x_0\rangle+\psi(x))\big\}$. For finite-sum forms of (1.1), $C$ is usually on the magnitude of $n\times C_1$.

**Algorithm 1** The AdaptReg Reduction

---

**Input:** an objective $F(\cdot)$ in Case 2 (smooth and not necessarily strongly convex);
$\quad\quad\quad$ $x_0$ a starting vector, $\sigma_0$ an initial regularization parameter, $T$ the number of epochs;
$\quad\quad\quad$ an algorithm $\mathcal{A}$ that solves Case 1 of problem (1.1).

**Output:** $\widehat{x}_T$.

1: $\widehat{x}_0 \leftarrow x_0$.
2: **for** $t \leftarrow 0$ **to** $T - 1$ **do**
3: $\quad\quad$ Define $F^{(\sigma_t)}(x) \stackrel{\text{def}}{=} \frac{\sigma_t}{2}\|x - x_0\| + F(x)$.
4: $\quad\quad$ $\widehat{x}_{t+1} \leftarrow \mathcal{A}(F^{(\sigma_t)}, \widehat{x}_t)$.
5: $\quad\quad$ $\sigma_{t+1} \leftarrow \sigma_t/2$.
6: **end for**
7: **return** $\widehat{x}_T$.

---

## 3 Related Works

Catalyst/APPA [11, 19] turn non-accelerated methods into accelerated ones, which is different from the purpose of this paper. They can be used as regularization reductions from Case 2 to Case 1 (but not from Cases 3 or 4); however, they suffer from two log-factor loss in the running time, and perform poor in practice [1]. In particular, Catalyst/APPA fix the regularization parameter throughout the algorithm but our AdaptReg decreases it exponentially. Their results cannot imply ours.

Beck and Teboulle [5] give a reduction from Case 4 to Case 2. Such a reduction does not suffer from a log-factor loss for almost trivial reason: by setting the smoothing parameter $\lambda = \varepsilon$ and applying any $O(1/\sqrt{\varepsilon})$-convergence method for Case 2, we immediately obtain an $O(1/\varepsilon)$ method for Case 4 without an extra log factor. Our main challenge in this paper is to provide log-free reductions to Case 1;[6] simple ideas fail to produce log-free reductions in this case because all efficient algorithms solving Case 1 (due to linear convergence) have a log factor. In addition, the Beck-Teboulle reduction is biased but ours is unbiased.

The so-called homotopy methods (e.g. methods with geometrically decreasing regularizer/smoothing weights) appeared a lot in the literature [6, 25, 31, 33]. However, to the best of our knowledge, all existing homotopy analysis either only work for specific algorithms [6, 25, 31] or solve only special problems [33]. In other words, none of them provides all-purpose black-box reductions like we do.

## 4 AdaptReg: Reduction from Case 2 to Case 1

We now focus on solving Case 2 of problem (1.1): that is, $f(\cdot)$ is $L$-smooth, but $\psi(\cdot)$ is not necessarily SC. We achieve so by reducing the problem to an algorithm $\mathcal{A}$ solving Case 1 that satisfies HOOD.

AdaptReg works as follows (see Algorithm 1). At the beginning of AdaptReg, we set $\widehat{x}_0$ to equal $x_0$, an arbitrary given starting vector. AdaptReg consists of $T$ epochs. At each epoch $t = 0, 1, \ldots, T-1$, we define a $\sigma_t$-strongly convex objective $F^{(\sigma_t)}(x) \stackrel{\text{def}}{=} \frac{\sigma_t}{2}\|x - x_0\|^2 + F(x)$. Here, the parameter $\sigma_{t+1} = \sigma_t/2$ for each $t \geq 0$ and $\sigma_0$ is an input parameter to AdaptReg that will be specified later. We run $\mathcal{A}$ on $F^{(\sigma_t)}(x)$ with starting vector $\widehat{x}_t$ in each epoch, and let the output be $\widehat{x}_{t+1}$. After all $T$ epochs are finished, AdaptReg simply outputs $\widehat{x}_T$.

We state our main theorem for AdaptReg below and prove it in the full version of this paper.

**Theorem 4.1** (AdaptReg)**.** *Suppose that in problem (1.1)* $f(\cdot)$ *is* $L$*-smooth. Let* $x_0$ *be a starting vector such that* $F(x_0) - F(x^*) \leq \Delta$ *and* $\|x_0 - x^*\|^2 \leq \Theta$. *Then,* AdaptReg *with* $\sigma_0 = \Delta/\Theta$ *and* $T = \log_2(\Delta/\varepsilon)$ *produces an output* $\widehat{x}_T$ *satisfying* $F(\widehat{x}_T) - \min_x F(x) \leq O(\varepsilon)$ *in a total running time of* $\sum_{t=0}^{T-1} \text{Time}(L, \sigma_0 \cdot 2^{-t})$.[7]

**Remark 4.2.** We compare the parameter tuning effort needed for AdaptReg against the classical regularization reduction. In the classical reduction, there are two parameters: $T$, the number of

iterations that does not need tuning; and $\sigma$, which had better equal $\varepsilon/\Theta$ which is an unknown quantity so requires tuning. In AdaptReg, we also need tune only one parameter, that is $\sigma_0$. Our $T$ need not be tuned because AdaptReg can be interrupted at any moment and $\widehat{x}_t$ of the current epoch can be outputted. In our experiments later, we spent the *same effort* turning $\sigma$ in the classical reduction and $\sigma_0$ in AdaptReg. As it can be easily seen from the plots, tuning $\sigma_0$ is much easier than $\sigma$.

**Corollary 4.3.** *When* AdaptReg *is applied to SVRG, we solve the finite-sum case of Case 2 with running time* $\sum_{t=0}^{T-1} \mathsf{Time}(L, \sigma_0 \cdot 2^{-t}) = \sum_{t=0}^{T-1} O(n + \frac{L2^t}{\sigma_0}) \cdot C_1 = O(n \log \frac{\Delta}{\varepsilon} + \frac{L\Theta}{\varepsilon}) \cdot C_1$. *This is faster than* $O\big(\big(n + \frac{L\Theta}{\varepsilon}\big) \log \frac{\Delta}{\varepsilon}\big) \cdot C_1$ *obtained through the old reduction, and faster than* $O\big(\frac{n + L\Theta}{\varepsilon}\big) \cdot C_1$ *obtained by SAGA [9] and SAG [27].*

*When* AdaptReg *is applied to* Katyusha, *we solve the finite-sum case of Case 2 with running time* $\sum_{t=0}^{T-1} \mathsf{Time}(L, \sigma_0 \cdot 2^{-t}) = \sum_{t=0}^{T-1} O(n + \frac{\sqrt{nL2^t}}{\sqrt{\sigma_0}}) \cdot C_1 = O(n \log \frac{\Delta}{\varepsilon} + \sqrt{nL\Theta/\varepsilon}) \cdot C_1$. *This is faster than* $O\big(\big(n + \sqrt{nL/\varepsilon}\big) \log \frac{\Delta}{\varepsilon}\big) \cdot C_1$ *obtained through the old reduction on* Katyusha *[1].*[8]

## 5  AdaptSmooth: **Reduction from Case 3 to 1**

We now focus on solving the finite-sum form of Case 3 for problem (1.1). Without loss of generality, we assume $\|a_i\| = 1$ for each $i \in [n]$ because otherwise one can scale $f_i$ accordingly. We solve this problem by reducing it to an oracle $\mathcal{A}$ which solves the finite-sum form of Case 1 and satisfies HOOD. Recall the following definition using Fenchel conjugate:[9]

**Definition 5.1.** *For each function* $f_i \colon \mathbb{R} \to \mathbb{R}$, *let* $f_i^*(\beta) \overset{\text{def}}{=} \max_\alpha \{\alpha \cdot \beta - f_i(\alpha)\}$ *be its Fenchel conjugate. Then, we define the following smoothed variant of* $f_i$ *parameterized by* $\lambda > 0$: $f_i^{(\lambda)}(\alpha) \overset{\text{def}}{=} \max_\beta \big\{\beta \cdot \alpha - f_i^*(\beta) - \frac{\lambda}{2}\beta^2\big\}$ . *Accordingly, we define* $F^{(\lambda)}(x) \overset{\text{def}}{=} \frac{1}{n} \sum_{i=1}^n f_i^{(\lambda)}(\langle a_i, x \rangle) + \psi(x)$ .

From the property of Fenchel conjugate (see for instance the textbook [28]), we know that $f_i^{(\lambda)}(\cdot)$ is a $(1/\lambda)$-smooth function and therefore the objective $F^{(\lambda)}(x)$ falls into the finite-sum form of Case 1 for problem (1.1) with smoothness parameter $L = 1/\lambda$.

Our AdaptSmooth works as follows (see pseudocode in the full version). At the beginning of AdaptSmooth, we set $\widehat{x}_0$ to equal $x_0$, an arbitrary given starting vector. AdaptSmooth consists of $T$ epochs. At each epoch $t = 0, 1, \dots, T - 1$, we define a $(1/\lambda_t)$-smooth objective $F^{(\lambda_t)}(x)$ using Definition 5.1. Here, the parameter $\lambda_{t+1} = \lambda_t/2$ for each $t \geq 0$ and $\lambda_0$ is an input parameter to AdaptSmooth that will be specified later. We run $\mathcal{A}$ on $F^{(\lambda_t)}(x)$ with starting vector $\widehat{x}_t$ in each epoch, and let the output be $\widehat{x}_{t+1}$. After all $T$ epochs are finished, AdaptSmooth outputs $\widehat{x}_T$.

We state our main theorem for AdaptSmooth below and prove it the full version of this paper.

**Theorem 5.2.** *Suppose that in problem (1.1),* $\psi(\cdot)$ *is* $\sigma$ *strongly convex and each* $f_i(\cdot)$ *is G-Lipschitz continuous. Let* $x_0$ *be a starting vector such that* $F(x_0) - F(x^*) \leq \Delta$. *Then,* AdaptSmooth *with* $\lambda_0 = \Delta/G^2$ *and* $T = \log_2(\Delta/\varepsilon)$ *produces an output* $\widehat{x}_T$ *satisfying* $F(\widehat{x}_T) - \min_x F(x) \leq O(\varepsilon)$ *in a total running time of* $\sum_{t=0}^{T-1} \mathsf{Time}(2^t/\lambda_0, \sigma)$.

**Remark 5.3.** We emphasize that AdaptSmooth requires less parameter tuning effort than the old reduction for the same reason as in Remark 4.2. Also, AdaptSmooth, when applied to Katyusha, provides the fastest running time on solving the Case 3 finite-sum form of (1.1), similar to Corollary 4.3.

## 6  JointAdaptRegSmooth: **From Case 4 to 1**

We show in the full version that AdaptReg and AdaptSmooth can work together to reduce the finite-sum form of Case 4 to Case 1. We call this reduction JointAdaptRegSmooth and it relies on a jointly exponentially decreasing sequence of $(\sigma_t, \lambda_t)$, where $\sigma_t$ is the weight of the convexity parameter that we add on top of $F(x)$, and $\lambda_t$ is the smoothing parameter that determines how we

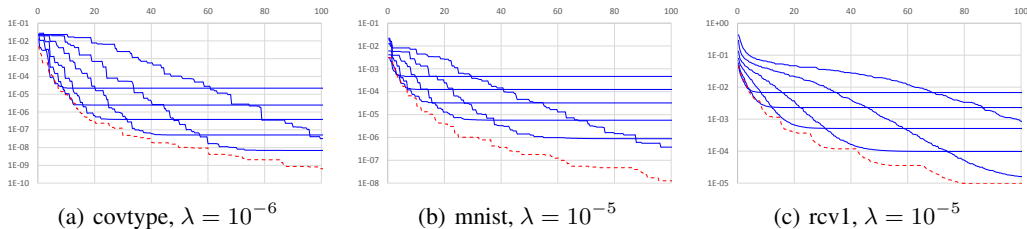

(a) covtype, $\lambda = 10^{-6}$      (b) mnist, $\lambda = 10^{-5}$      (c) rcv1, $\lambda = 10^{-5}$

Figure 1: Comparing AdaptReg and the classical reduction on Lasso (with $\ell_1$ regularizer weight $\lambda$). $y$-axis is the objective distance to minimum, and $x$-axis is the number of passes to the dataset. The blue solid curves represent APCG under the old regularization reduction, and the red dashed curve represents APCG under AdaptReg. For other values of $\lambda$, or the results on SDCA, please refer to the full version of this paper.

change each $f_i(\cdot)$. The analysis is analogous to a careful combination of the proofs for AdaptReg and AdaptSmooth.

# 7 Experiments

We perform experiments to confirm our theoretical speed-ups obtained for AdaptSmooth and AdaptReg. We work on minimizing Lasso and SVM objectives for the following three well-known datasets that can be found on the LibSVM website [10]: covtype, mnist, and rcv1. We defer some dataset and implementation details the full version of this paper.

## 7.1 Experiments on AdaptReg

To test the performance of AdaptReg, consider the Lasso objective which is in Case 2 (i.e. non-SC but smooth). We apply AdaptReg to reduce it to Case 1 and apply either APCG [20], an accelerated method, or (Prox-)SDCA [29, 30], a non-accelerated method. Let us make a few remarks:

- APCG and SDCA are both *indirect* solvers for non-strongly convex objectives and therefore regularization is intrinsically required in order to run them for Lasso or more generally Case 2.
- APCG and SDCA do not satisfy HOOD in theory. However, they still benefit from AdaptReg as we shall see, demonstrating the practical value of AdaptReg.

**A Practical Implementation.** In principle, one can implement AdaptReg by setting the termination criteria of the oracle in the inner loop as precisely suggested by the theory, such as setting the number of iterations for SDCA to be exactly $T = O(n + \frac{L}{\sigma_t})$ in the $t$-th epoch. However, in practice, it is more desirable to automatically terminate the oracle whenever the objective distance to the minimum has been sufficiently decreased. In all of our experiments, we simply compute the duality gap and terminate the oracle whenever the duality gap is below $1/4$ times the last recorded duality gap of the previous epoch. For details see the full version of this paper.

**Experimental Results.** For each dataset, we consider three different magnitudes of regularization weights for the $\ell_1$ regularizer in the Lasso objective. This totals 9 analysis tasks for each algorithm.

For each such a task, we first implement the old reduction by adding an additional $\frac{\sigma}{2}\|x\|^2$ term to the Lasso objective and then apply APCG or SDCA. We consider values of $\sigma$ in the set $\{10^k, 3 \cdot 10^k : k \in \mathbb{Z}\}$ and show the most representative six of them in the plots (blue solid curves in Figure 3 and Figure 4). Naturally, for a larger value of $\sigma$ the old reduction converges faster but to a point that is *farther* from the exact minimizer because of the bias. We implement AdaptReg where we choose the initial parameter $\sigma_0$ *also* from the set $\{10^k, 3 \cdot 10^k : k \in \mathbb{Z}\}$ and present the best one in each of 18 plots (red dashed curves in Figure 3 and Figure 4). Due to space limitations, we provide only 3 of the 18 plots for medium-sized $\lambda$ in the main body of this paper (see Figure 1), and include Figure 3 and 4 only in the full version of this paper.

It is clear from our experiments that

- AdaptReg is more efficient than the old regularization reduction;
- AdaptReg requires no more parameter tuning than the classical reduction;

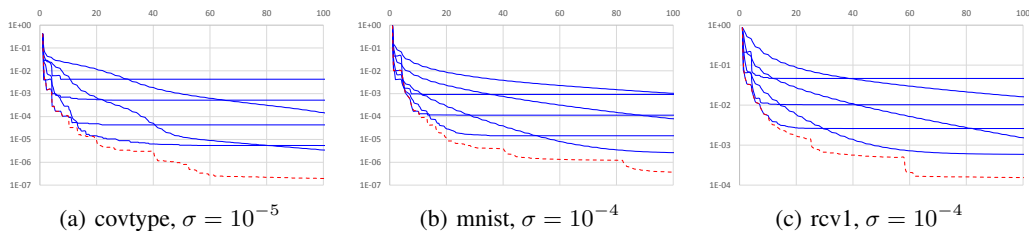

(a) covtype, $\sigma = 10^{-5}$        (b) mnist, $\sigma = 10^{-4}$        (c) rcv1, $\sigma = 10^{-4}$

Figure 2: Comparing AdaptSmooth and the classical reduction on SVM (with $\ell_2$ regularizer weight $\lambda$). $y$-axis is the objective distance to minimum, and $x$-axis is the number of passes to the dataset. The blue solid curves represent SVRG under the old smoothing reduction, and the red dashed curve represents SVRG under AdaptSmooth. For other values of $\sigma$, please refer to the full version.

- AdaptReg is unbiased so simplifies the parameter selection procedure.[10]

### 7.2   Experiments on AdaptSmooth

To test the performance of AdaptSmooth, consider the SVM objective which is non-smooth but SC. We apply AdaptSmooth to reduce it to Case 1 and apply SVRG [14]. We emphasize that SVRG is an *indirect* solver for non-smooth objectives and therefore regularization is intrinsically required in order to run SVRG for SVM or more generally for Case 3.

**A Practical Implementation.**   In principle, one can implement AdaptSmooth by setting the termination criteria of the oracle in the inner loop as precisely suggested by the theory, such as setting the number of iterations for SVRG to be exactly $T = O(n + \frac{1}{\sigma \lambda_t})$ in the $t$-th epoch. In practice, however, it is more desirable to automatically terminate the oracle whenever the objective distance to the minimum has been sufficiently decreased. In all of our experiments, we simply compute the Euclidean norm of the full gradient of the objective, and terminate the oracle whenever the norm is below $1/3$ times the last recorded Euclidean norm of the previous epoch. For details see full version.

**Experimental Results.**   For each dataset, we consider three different magnitudes of regularization weights for the $\ell_2$ regularizer in the SVM objective. This totals 9 analysis tasks. For each such a task, we first implement the old reduction by smoothing the hinge loss functions (using Definition 5.1) with parameter $\lambda > 0$ and then apply SVRG. We consider different values of $\lambda$ in the set $\{10^k, 3 \cdot 10^k : k \in \mathbb{Z}\}$ and show the most representative six of them in the plots (blue solid curves in Figure 5). Naturally, for a larger $\lambda$, the old reduction converges faster but to a point that is *farther* from the exact minimizer due to its bias. We then implement AdaptSmooth where we choose the initial smoothing parameter $\lambda_0$ also from the set $\{10^k, 3 \cdot 10^k : k \in \mathbb{Z}\}$ and present the best one in each of the 9 plots (red dashed curves in Figure 5). Due to space limitations, we provide only 3 of the 9 plots for small-sized $\sigma$ in the main body of this paper (see Figure 2, and include Figure 5 only in full version.

It is clear from our experiments that

- AdaptSmooth is more efficient than the old smoothing reduction, especially when the desired training error is small;
- AdaptSmooth requires no more parameter tuning than the classical reduction;
- AdaptSmooth is unbiased and simplifies the parameter selection for the same reason as Footnote 10.

## Footnotes

[2]Recall that obtaining the *optimal* convergence rate is one of the main goals in operations research and machine learning. For instance, obtaining the optimal $1/\varepsilon$ rate for online learning was a major breakthrough since the $\log(1/\varepsilon)/\varepsilon$ rate was discovered [13, 15, 26].

[3]Smoothing reduction is typically applied to the finite sum form only. This is because, for a general high dimensional function $f(x)$, its smoothed variant $\widehat{f}(x)$ may not be efficiently computable.

[4]More formally, one needs this variant to satisfy $|\widehat{f}_i(\alpha) - f_i(\alpha)| \le \varepsilon$ for all $\alpha$ and be smooth at the same time. This can be done at least in two classical ways if $\widehat{f}_i(\alpha)$ is Lipschitz continuous. One is to define $\widehat{f}_i(\alpha) = \mathbb{E}_{v \in [-1,1]}[f_i(\alpha + \varepsilon v)]$ as an integral of $f$ over the scaled unit interval, see for instance Chapter 2.3 of [12], and the other is to define $\widehat{f}_i(\alpha) = \max_\beta \left\{ \beta \cdot \alpha - f_i^*(\beta) - \frac{\varepsilon}{2}\alpha^2 \right\}$ using the Fenchel dual $f_i^*(\beta)$ of $f_i(\alpha)$, see for instance [24].

[5]Although our definition is only for deterministic algorithms, if the guarantee is probabilistic, i.e., $\mathbb{E}\big[F(x')\big]-\min_x F(x)\leq\frac{F(x_0)-\min_x F(x)}{4}$, all the results of this paper remain true. Also, the constant 4 is very arbitrary and can be replaced with any other constant bigger than 1.

[6]Designing reductions to Case 1 (rather than for instance Case 2) is crucial for various reasons. First, algorithm design for Case 1 is usually easier (esp. in stochastic settings). Second, Case 3 can only be reduced to Case 1 but not Case 2. Third, lower bound results [32] require reductions to Case 1.

[7]If the HOOD property is only satisfied probabilistically as per Footnote 5, our error guarantee becomes probabilistic, i.e., $\mathbb{E}\big[F(\widehat{x}_T)\big] - \min_x F(x) \leq O(\varepsilon)$. This is also true for other reduction theorems of this paper.

[8]If the old reduction is applied on APCG, SPDC, or AccSDCA rather than Katyusha, then two log factors will be lost.

[9]For every explicitly given $f_i(\cdot)$, this Fenchel conjugate can be symbolically computed and fed into the algorithm. This pre-process is needed for nearly all known algorithms in order for them to apply to non-smooth settings (such as SVRG, SAGA, SPDC, APCG, SDCA, etc.).

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
