[Reviews · NeurIPS 2016]

Reviewer 1

Summary

The authors consider the composite convex optimization problem of minizing the sum of a smooth and a simple convex function, which prox operator is efficiently computatble. The analysis of the authors is based on an abstract decrease property of descent methods for smooth strongly convex optimization: the HOOD property. The main mechanisms behind the proposed method is to approximate convex problems by smooth strongly convex models and tune the level of approximation as a function of the iteration counter. The authors propose and analyse nested algorithms, the inner loop is a call to an abstract descent method which satisfies the HOOD property and the outer loop consists in adjusting the smoothness and / or strong convexity levels of approximate objective function. Careful tuning of approximate model parameters and inner loop calls to HOOD algorithms allows the authors to derive convergence rates for the proposed reductions. Combining these results with known methods for strongly convex optimization, the authors derive algorithms with optimal convergence rates in important optimization model classes. The results are illustrated on Lasso and SVM problems with benchmark datasets.

Qualitative Assessment

Overall, the paper is convincing and well written. The problem is very relevant for machine learning applications and the proof arguments are quite convincing. I could not find major flaw in the paper. I have several concerns to raise regarding this work. - Presentation and comparison with existing methods of similar flavor is not really fair. Smoothing has a long history in optimization and recent work on accelerated methods through strong convexity are not properly presented. Some elements of discussion are missing and no concurrent algorithm is implemented in the numerical section. - The authors put a lot of emphasis on what they call the "old" or "classic" reduction methods. There is no reference about these methods and related complexity results mentioned in the paper. Furthermore, most of the discussion about comparison with state of the art focuses on these "old" reductions. I am not really convinced that this is a fair baseline. - Some aspects of the discussion regarding lasso are very inaccurate. %% Details about the main comment Comparison with existing literature: Smoothing nonsmooth objectives is a relevant topic in optimization literature. For example reference [20] in the main text or the following reference Beck, Amir, and Marc Teboulle. "Smoothing and first order methods: A unified framework." SIAM Journal on Optimization 22.2 (2012): 557-580. I guess the superiority of the proposed framework is the "anytime" property, but this deserved to be presented more clearly. The authors present the work in [8] and [15] as being identical to the "traditional regularization reduction". I do not understand this point. These works are based on extensions of the proximal point algorithm which indeed adds a strongly convex term with a fixed modulus of strong convexity at each step. But the term itself depends on the iterate counter which is very different from what is described as the classical reduction method. These methods may suffer from log factors, but I disagree with the fact that they suffer from bias. There is no discussion about parameters that are required for the proposed algorithm compared to these methods. For example, in Theorem 3.1, AdaptReg requires an estimate of the distance to the optimum point. How to estimate this? Is the method robust to under estimation of this parameter? Do concurrent methods require this parameter. To my understanding, Catalyst only requires a bound on suboptimality which is easier to estimate. Similar comments hold for Theorem C.2. Finally, no concurrent or even baseline method is implemented apart from the "old" reduction which I think weakens considerably the numerical illustration. Indeed, they mostly illustrate the fact that the "old" reduction is not an anytime method. The complexity results regarding the old reduction method in section 1.1 should be justified by proper references. %% Additional questions and suggestions The bibliography should be in alphabetical order. Ref [27] is not cited in the main text. On line 24 it is argued that "for a non-SC objective such as Lasso, its dual is not even be well-defined". First, there is a typo and, second, this does not make sense to me. The dual is well defined, feasible and its optimal value is attained. On page 3, the authors mention optimality of the convergence rates they obtain and justify them by citing reference [1]. From my understanding, the mentioned paper does not focus on lower bounds and there should be a better reference. In remark 3.2, the authors mention that it is much easier to tune \sigma_0 than \sigma because this is "seen from the plots". This remark is repeated in the numerical section and I do not find that this discussion is convincing. The presentation of Section 6.1 is very fuzzy. Contrary to what is said, the lasso problem is not smooth. There must be a confusion somewhere. What is meant by indirect solver on line 246? The whole analysis is based on the HOOD property and the numerical illustration is done with methods which do not satisfy this property. This is a bit disappointing. Since the duality gap is only an upper bound on suboptimality, the remark on line 255 is only an heuristic and is not sufficient to ensure that the results of the paper apply. This could be precised. A similar comment holds for line 283. The numerical section does not describe what happens for a bad choice of \sigma_0. At the end of appendix A, it is mentioned that it is not possible to compute duality gaps for the Lasso. I know at least one possibility from the Appendix of Julien Mairal's Thesis. Moreover, the norm of the gradient is not really the best indicator of suboptimality since it is not 0 in general at the optimum for the lasso. May be the authors refer to the gradient mapping (which includes the prox step). Or may be this is related to the confusion that I pointed out earlier.

Confidence in this Review

2-Confident (read it all; understood it all reasonably well)


Reviewer 2

Summary

The problem of composite optimization where the objective F(x) is the sum of regularizer g(x) plus a loss function f(x) comes in various applications in machine learning. Depending on the application, the objective function may possess different convexity and smoothness properties. There are mainly four categories: 1) g is strongly convex and f is smooth. 2) g is non-strongly convex and f is smooth 3) g is strongly convex and f is non-smooth 4) g is non-strongly convex and f is non-smooth There are plenty of algorithms that are applicable to only some of the four categories above. Therefore, it is interesting to be able to develop a generic framework where one takes a method developed for one of the categories as a black box and make it applicable to others. Such methods are called reduction techniques. In this paper, the authors develop a new optimal and practical reduction method where one can save some extra log factors compared to existing reduction strategies (such as classical reduction, Catalyst or APPA) without introducing a bias in the objective value. Numerical experiments are carried out to show that the efficiency of the reduction method. The method is novel to my knowledge and makes some of the ideas leading to reduction more transparent. Some questions and comments: 1) Quoting from the paper, line 34: " after applying existing reductions, algorithms become biased so the objective value does not converge to the optimal solution"... Catalyst is an existing reduction method as you mention in the paper. Does not the objective value converge to the global minimum for the Catalyst method? 2) Line 61: In the expectation formula that defines the smoothed variant, the definition of the variable v is not clear. It seems that v is a random variable uniformly distributed over [-1,1], although it was not entirely clear to me. 3) Line 76, definition of the HOOD property: I was wondering, whether there is any particular reason for choosing the constant 4 in the definition. Would the constant 2 work? I would suggest to add a remark on this choice.

Qualitative Assessment

The paper is well-written. I believe the approach makes some of the ideas behind reduction techniques more transparent.

Confidence in this Review

2-Confident (read it all; understood it all reasonably well)


Reviewer 3

Summary

The paper proposes an optimization framework called AdaptReg Reduction to solve a class of non-strongly composite convex minimization problems. The authors introduce a new concept called HOOD and provide an algorithm framework that unifies many existing algorithms but is applicable to non-strongly convex functions. They estimate the complexity of their algorithm using the HOOD concept. A numerical test on sparse logistic regression is provided to illustrate the performance of this algorithm compared to APCG.

Qualitative Assessment

I think the paper presents a good practical idea, which allows one to solve non-strongly convex problems using existing algorithms for strongly convex problems including SVRG, and dual first-order methods. The adaptive rule for strong convexity parameter is clearly important to accelerate the actual performance of the algorithm. The idea of such a technique is unfortunately covered by some existing papers, for example, A Universal Catalyst for First-Order Optimization, NIPS 2015. In addition, the reviewer still sees many points that should be discussed from this paper. - First, the HOOD definition seems to be too restricted, which is only satisfied in two cases: The condition number of the problem is ideal, near 4/3 or 1/sqrt(3/4), or the algorithm A must be performed up to a large number of iterations to reach this condition. - Second, for non-strongly convex problems, the condition number of the subproblem solved by A depends on sigma_t, which is decreasing at a linear rate (by a bisection rule) and can be very small soon. In this case, the condition number of the subproblem is huge and A requires many iterations to reach the HOOD condition. This is somehow not natural in homotopy approach where the parameter is probably decreasing slowly to keep the number of iterations in A low. - Third, for the dual approach, I have a feeling that the following two papers may have a connection to this paper. http://link.springer.com/article/10.1007%2Fs11750-014-0326-z#page-1 http://arxiv.org/abs/1509.00106

Confidence in this Review

2-Confident (read it all; understood it all reasonably well)


Reviewer 4

Summary

This paper describes new reductions of Lipschitz convex optimizations to the smooth, strongly convex case. The reduction is performed by constructing a smoothed, regularized objective and iteratively optimizing this objective with a warm start from the previous result while geometrically decreasing the smoothing and regularization terms. Experimental comparisons to SVRG and SDCA are favorable.

Qualitative Assessment

The results are clearly presented. But it is less clear to me how this result extends prior work. The authors briefly mention Catalyst and APPA in the related works section at the bottom of page 3. But they dismiss these approaches, arguing that they “continue to introduce bias and suffer from a log factor loss in running time.” My understanding of APPA is that by “un-regularizing” it achieves an unbiased reduction. Could the authors clarify how this work distinguishes itself?

Confidence in this Review

1-Less confident (might not have understood significant parts)


Reviewer 5

Summary

In this paper, the authors applied (primal or dual) smoothing techniques to approximate a convex objective which is the sum of a possibly non-strongly convex function and a possibly nonsmooth function by a sequence of functions that are the sum of a strongly convex function and a smooth function. Optimal methods are then applied to solving the later problems successively up to a certain accuracy, resulting in (claimed) faster convergence rates than those known in the literature. The reduction technique is similar to those widely used "homotopy" or "continuation" method.

Qualitative Assessment

The overall presentation is clear. I have three major pieces of comments. Part 1: As mentioned above, the technique described is essentially the homotopy method or continuation technique. I do not see any references on this. The authors should look up the literature and comment on those methods. The following reference is a good starting point: I. L. Xiao and T. Zhang. A proximal-gradient homotopy method for the sparse least-squares problem. SIAM J. Optim. 23(2), 1062-1091 (2013). Indeed, comparing with those homotopy methods, the main contribution in this work seems to be the introduction of the HOOD property, and then explicitly writing down the resulting complexity. Since NIPS is a top conference, it is important for the papers to address clearly existing/related literature. Part 2: Some errors in mathematics/some claims that need clarifications: I. Line 24: it is wrong to say that the dual of LASSO is not well defined. Of course it is well defined, and is a constrained optimization problem. II. Line 56: I think using accelerated gradient method to F (batch method) one can also obtain O(sqrt(L/epsilon)). So, what is the point of the continuation/homotopy technique? III. Line 65: Same as the above: using batch accelerated gradient method to the dual, one can also obtain O(sqrt(1/sigma/epsilon)). So, why the suggested approach? I guess the authors might be focusing on the stochastic/parallel settings in this paper. If so, please modify these claims accordingly. In particular, please emphasize that this paper is for stochastic/parallel settings right AT THE BEGINNING. IV. In the definition of HOOD: why 1/4? V. Section 6.1: It is more or less well known that by smoothing out the L1 regularizer, one does not get sparse iterates anymore and some manual truncation needs to be performed to obtain sparse solutions. How does this affect the quality of solution, comparing with direct methods? Does this result in extra iterations in order to obtain a solution with desired quality? VI. Line 383 in the appendix: there are several tricks for computing duality gap for the LASSO problem, see for example, Section 4.2 of the following: "B. Wen, X. Chen and T. K. Pong. Linear convergence of proximal gradient algorithm with extrapolation for a class of nonconvex nonsmooth minimization problems. arXiv preprint arXiv: 1512.09302v1." Part 3: Minor typos: I. Line 441 of appendix: - lambda_t G^2/2 instead of plus.

Confidence in this Review

2-Confident (read it all; understood it all reasonably well)